# Prevalence of Undernutrition and Anemia among Santal Adivasi Children, Birbhum District, West Bengal, India

**DOI:** 10.3390/ijerph17010342

**Published:** 2020-01-03

**Authors:** Caroline Katharina Stiller, Silvia Konstanze Ellen Golembiewski, Monika Golembiewski, Srikanta Mondal, Hans-Konrad Biesalski, Veronika Scherbaum

**Affiliations:** 1Institute of Nutritional Sciences, University of Hohenheim, 70593 Stuttgart, Germany; 2Shining Eyes -medical aid for children and socioeconomic village development in India e.V., 74223 Flein, Germany; 3Institute of Household and Consumer Economics (530A), University of Hohenheim, 70599 Stuttgart, Germany; 4Bolpur Manab Jamin, South Jambuni, Bolpur, Birbhum District, West Bengal 731204, India

**Keywords:** young child undernutrition, CIAF, morbidity, anemia, independent predictors of anemia, socio-economic and demographic information, Santals, Adivasi, West Bengal, India

## Abstract

India’s Adivasi scheduled tribe population is disproportionately affected by undernutrition and anemia, thereby prevailing in the poorest wealth deciles denominated as socially and economically vulnerable. This study was designed to assess the extent of child undernutrition (conventional and composite index of anthropometric failure (CIAF) classification), as well as the burden of anemia in children and its independent nutrition specific and sensitive drivers, moreover to reflect the living conditions of Santal Adivasis. The research survey was conducted in 21 Santal villages, Birbhum District, West Bengal, in 2015. An overall 307 children (aged 6–39 months) and their mothers (*n* = 288) were assessed for their hemoglobin (Hb) levels (HemoCue Hb201+) and anthropometric indices such as height/length, weight and mid-upper arm circumference (MUAC). Moreover, socio-demographic household characteristics were surveyed. The study confirmed Adivasi children lagging behind national average with a high prevalence of undernutrition (height-for-age z-score (HAZ) 51.9%, weight-for-age z-score (WAZ) 49.2%, weight-for-height z-score WHZ 19.0% and CIAF 61.6%) and of moderate and severe anemia (Hb < 10 g/dL, 73.3% altogether). Child’s age <24 months, low WAZ scores, morbidity (any fever, diarrhea or respiratory infection) on the checkup day or during previous week, low maternal Hb level, and lack of dietary diversification were identified as predictors for anemia, thereby warrant targeted interventions to decrease the high anemia rates assessed in the study site.

## 1. Introduction

The term ‘Scheduled Tribes’ first appeared in the Constitution of India. Article 366 (25) defines scheduled tribes as: “such tribes or tribal communities or parts of or groups within such tribes or tribal communities as are deemed under Article 342 to be Scheduled Tribes for the purposes of this constitution“ [1]. Scheduled tribes in India account for 8.6% of the total population as per the Census conducted in 2011 [2]. These tribal people are commonly called “Adivasis”—an umbrella term for diverse ethnic and tribal groups considered as India’s indigenous population. In total, there are 573 communities being recognized as scheduled tribes [3], where Santals form the third largest Adivasi community, constituting half of all scheduled tribes in West Bengal (52%), with Oraon (14.0%), Munda (7.8%), Bhumij (7.6%) and Kora (3.2%) being other major tribes. All tribes predominantly reside in the rural areas (93.9%) [4]. While national poverty rates have diminished by 20% in the last two decades, Adivasis constitute one fourth of the population being counted to the poorest wealth decile [5]. Santals residing at Birbhum District were found to live on very low incomes, earning their living largely in agriculture, often as day laborers or with minor landholdings [6]. To complete the picture, tribal children are known to lag far behind in school attainments above the primary level. Further a smaller proportion of tribal children receives qualified medical treatment linked to a physical remoteness to health facilities but also due to deep-rooted cultural mistrust between conventional medicine providers and Adivasi people [7]. Under-five-mortality rates among Adivasis are by 25% increased as compared to non-Adivasi children aged younger than five years. There is also a rise in severe wasting and severe stunting among Adivasi infants persisting throughout early childhood [5]. This reality is also reflected in the data of the National Family Health Survey-4 on children below five years of age with scheduled tribes [8] showing worse levels of undernutrition as compared to average numbers of India [9]: height-for-age z-score (HAZ; 43.8% vs. 38.4%), weight-for-age z-score (WAZ; 45.3% vs. 35.8%), weight-for-height z-score (WHZ; 27.4% vs. 21.0%) or anemia rates (63.3% vs. 58.5%).

Child undernutrition and poor child development is the outcome of immediate causes like inadequate dietary intake or health status (nutrition-specific drivers), as well as underlying causes including the lack of access to adequate food, appropriate care or adequate health services (nutrition-sensitive drivers) and basic causes that contradict an enabling/healthy environment rooted in political superstructure and human, economic and organizational resources, as conceptualized by the UNICEF’s framework of child’s nutrition, health, survival and development [10,11,12]. The major objective of this research work was to elucidate the burden of undernutrition and anemia in Santal children (descriptive analysis). Secondary objectives were to define independent nutrition-specific and sensitive predictors of moderate and severe anemia (Hb < 10 g/dL), as well as to reflect the living conditions of the Santal communities, Birbhum District, West Bengal.

## 2. Materials and Methods

### 2.1. Village Selection, Household Survey and Medical Checkup

Presented baseline data were obtained in the scope of a prospective longitudinal feeding trial lasting for 1.5 years, initiated by the registered association Shining Eyes e.V. and implemented in cooperation with the non-governmental organization (NGO) Bolpur Manab Jamin. All 21 tribal villages in Bolpur Manab Jamin’s sphere of activity were approached to be included in the research (the detailed study design and effect analysis is elaborated elsewhere at a later stage, cluster-randomization was applied in order to allocate the villages to the respective intervention arms).

Socio-demographic characteristics were obtained retrospectively in terms of a baseline household questionnaire conducted between Dec 2014 and Apr 2015. For survey data collection a pre-tested semi-structured questionnaire was applied (the pre-testing was conducted in ten Santal households in neighboring villages to the study site). The questionnaire was designed based on a set of sample questionnaires that had been successfully applied during field research in developing countries, and was further supplemented with specific questions relevant to the current research and optimized based on the long-lasting experience of project coordinators in the study area. The interviews were individually scheduled by 11 trained social workers (who had received a three-day-training by research authorities as well as by the field-coordinator of the NGO Manab Jamin). To draw a holistic picture on the living circumstances of tribal families information was obtained regarding literacy status, agricultural assets, livelihood, hygienic habits, morbidity pattern of the child and health seeking behavior including accessibility to health care or sanitary facilities. Moreover the household structure and socio-economic and environmental conditions were elicited.

The 1st medical checkup in February 2015 with subsequent enrolment of participants provided baseline data on anthropometry, hemoglobin levels (Hb) and visible signs of undernutrition, dental disorders, skin infections, intestinal or respiratory infections based on the caretakers report and the doctor’s diagnosis; also heart defects or handicaps were identified.

All data obtained during baseline assessment (household survey and 1st medical checkup) constitute the concern of this paper (a recently submitted paper elaborates the topics maternal anthropometric data, child spacing, child caring and infant feeding practices also assessed during baseline).

### 2.2. Target Group and Response Rate

Children aged 6–36 months at baseline assessment were the study’s primary target group. At baseline assessment a total of 307 children (aged 6–39 months) were analyzed, 143 (46.6%) girls and 164 boys (53.4%). Hereby, thirteen children aged similar to the proposed age range with 37 up to 39 months, were decided to be kept in the baseline assessment due to limited sample size and as no bias is presumed. Three children initially measured were subsequently excluded from further analysis as the children were aged younger or considerably older than the proposed age-range.

All people of the included villages are beneficiaries of existing government schemes by the same means.

A total of 1692 families resided in the project area (21 villages), thereof 562 individuals were potential beneficiaries of the trial (i.e., children in the relevant age range, pregnant or lactating mothers. Note: the intervention trial also entitled pregnant and lactating mothers to benefit from medical checkups or the feeding program, in order to cover the 1000 days of the window of opportunity). When checked for tribal affiliation 503 individuals remained (59 have been identified as non-tribal). Thereof, merely one rejection was noted in the end of 2014. Altogether 335 mother-child pairs, (not considering pregnant and lactating women) were eligible for the study and associated measures (household survey and medical checkups). However merely 310 children were finally enrolled at the baseline assessment (the remaining ones were not available in the villages in February 2015, e.g., moved away permanently, resided for some time at relative places, or due to unknown reason), with 307 being included in the final analysis.

In regard to the baseline household survey, 291 caretakers participated (response rate 98% of *n* = 297 approached households total). The main caretaker was in 98.6% of cases the mother. Ten mothers had two children participating in the study (nine mothers had two singleton children, one mother had twins).

The high participation rate of 99.8% in the trial, and similarly high response rate of 98% in the household survey may be explained by the commonly grateful acceptance of, e.g., government nutrition programs amongst villagers in the District area. Villages cluster-randomized to the control group (receiving no nutrition intervention) were equally beneficiaries of medical checkups. Independent of study participation, to all residents of the 21 villages, kitchen garden and awareness programs were offered during the follow-up phase after completion of the trial. Still, the disclosure of these awaiting preventive measures may have increased the willingness of participants to attend medical measurements or the time-consuming household survey.

### 2.3. Sample Size Calculation

Estimating the population proportion with specified absolute precision resulted in a minimum required sample size of *n* = 288 [13]. Hereby the confidence level was set at 95%, the margin of error at 0.05, and the estimated prevalence of any anemia in the age group 6–35 months for scheduled tribes accounted for 75.3%. The basis for this percentage were data obtained from the NFHS-4, West Bengal, 2015–2016 [14].

Testing the overall fit of the regression model requires a minimum acceptable sample size of *n* = 138, following the proposed rule of thumb 50 + 8 m (with m being the number of predictors) [15,16].

### 2.4. Outcome Variables

Hb level was determined by using a portable hemoglobinometer (HemoCue Hb 201+).

Mid-upper arm circumference (MUAC) was measured to the nearest millimeter at the mid-point of the left arm (relaxed position), using a non-stretchable tape. All subjects were weighed to the accuracy of 0.1 kg by a calibrated seca-floor-scale with 2 in 1 (mother-child function), wearing no shoes and minimal clothes (shirt and shorts for children). Height was measured for children ≥2 years by using seca-mobile stadiometer at the accuracy of 0.1 cm. Length measurement was attempted for children <2 years and was performed by using seca-mobile-mat at the accuracy of 0.5 cm.

The child’s date of birth was obtained from the birth certificate.

In this study the World Health Organization (WHO) Anthro software 2011 [17] served to compute the stunting, underweight, wasting, body mass index (BMI)-for-age and mid-upper arm circumference (HAZ, WAZ, WHZ, BAZ and MUAC) scores with the 2006 WHO child growth standard as reference [18]. The interpretation of MUAC (cm) or BMI was performed according to cutoffs adopted by the WHO and joint United Nations (UN) statement [19], similarly for Hb levels cutoffs by WHO were applied [20].

### 2.5. Statistics

The data were analyzed using SPSS statistics 22 (IBM, Armonk, NY, USA). The general level of significance was set at *p* < 0.05. The results are expressed in percentages or in means ± SD/median. Tests applied were the Kolmogorov–Smirnov Test, the student’s *t*-test/Mann–Whitney test and univariate ANOVA/Kruskal–Wallis test. In order to ascertain amongst which groups a significant difference was found, the Duncan post-hoc test was performed or in case of non-normal data the Mann–Whitney test with subsequent correction of the alpha inflation (cumulative Type I error) according to Bonferroni. To elucidate associations between two continuous variables, the Pearson/Spearman correlation was applied. Sets of categorical data were assessed by Pearson’s chi-square test, or Fisher’s exact test.

As the prevalence of any anemia was very high during baseline assessment (93.8%) with merely 6.2% (*n* = 19) being non-anemic, predictors for the positive outcome of being non-anemic/mildly anemic (26.7%, *n* = 82) were investigated. During analysis the cutoff Hb ≥ 10.0 g/dL (no anemia and mild anemia) was chosen, in order to achieve a reasonable sample size during regression analysis and as the condition of mild anemia is not of priority for medical treatment in India, with health screenings in children being targeted at moderate and severe forms of anemia. Modeling non-anemia/mild anemia in children was performed using the commonly used binary logistic regression. In cases when the outcome is not rare (outcome >10%), odds ratio (OR) is no longer a good approximation of the risk (RR) or prevalence ratio (PR). Hence in common conditions, the OR can merely be interpreted in terms of ratio of odds as measure of association [21]. Still, if needed from the OR a risk ratio can be approximated by applying conversion formulas for OR’s [22,23,24]. Yet, as long as ORs are interpreted as such, there is no need for any transformation, which anyhow may be challenging in particular for continuous covariates [23]. Moreover not the exact strength of association is the major focus of this research, rather to find out any predictors favoring non-anemia/mild anemia in the child (forward-Wald regression model). Thus, the PR which is discussed as the measure of choice for studies with common outcomes [22,25,26], has not been assessed. Even though the interpretation of odds may be not as intuitive as of RRs, Cook advocates the use of ORs in particular in studies in which outcomes are common [27]. ORs have some properties preferable to RRs (homogeneity allows reliable application to all individuals within a population, show symmetrical behavior with outcome definition, etc.) [24,27].

### 2.6. Predictor Analysis

In order to assess the independent effects on non-anemia/mild anemia, a list of potential predictors was developed, guided by the UNICEF conceptual framework of malnutrition causation. Hereby crucial covariates of the subcategories mother and child factors, nutritional factors and socio-economic food security factors were elucidated according availability from the survey. First each potential predictor variable (about 35 variables) was examined for the presence of statistical association with the dependent variable by binary regression (generation of crude *p*-values).

In a second step, in case of related variables the most relevant one was selected on the basis of statistical significance and informational content provided. To enter the multivariable logistic regression a *p*-value of <0.25 in the bivariate analysis was the overarching criteria [28,29,30]. Merely the maternal educational level, well-recognized as a nutrition-sensitive indicator, was included independently of its significance level in the logistic regression model with a forward stepwise approach. Independent predictors of Hb ≥10 g/dL were thereby identified. The following 11 covariates were enclosed in the initial model: children’s age, morbidity history at medical checkup or during previous week, heart defect, educational level of the mother, household cash income (categorical variables, for reference categories (also see Table 7): ≥24 months, no morbidity, any heart defect, any formal education, ≥5000 Indian Rupee (INR)/month, respectively), as well as WAZ, maternal Hb, maternal BMI, count of food groups consumed the day before the interview, delivery number/birth order of study child and caring time (continuous variables). The rationale for the cutoffs used in the case of categorical variables was based on the design of survey questions. For age the cutoff 24 months was applied, as at this point of time an increase in Hb levels was noted (trend line in scatterplot matrix) and as the continuous variable showed interactions with their logit odds.

All continuous predictors were assessed for interactions with their logit (log odds), and all relationships were found to be linear (Box–Tidwell test). Further the initial model was assessed for the presence of multi-collinearity [31], interaction (forward regression) and confounding (enter regression, not presented in this manuscript). Fitness to the data was proofed by the Hosmer–Lemeshow’s goodness-of-fit test.

Finally variables with a *p*-value below 0.05 in the multivariable analysis were declared to be significantly associated with the dependent variable Hb ≥10 g/dL, and retained in the final model. These variables are presented in terms of standard errors, ORs and 95% CI, as obtained by the forward regression.

### 2.7. Trial Registration

The trial was retrospectively registered at the German Clinical Trials Register on the 1st July 2019 (DRKS00017388).

### 2.8. Ethics Approval and Consent to Participate

The intervention trial is registered with the study code: 014/1763 at the Freiburg Ethics Commission International (10th Nov 2014), and approved by the local Child Development Office in Bolpur, Birbhum District (11th July 2014). Before enrolment, a full explanation of the study purpose was given to the communities, and informed consent was obtained either by signature or thumbprint.

## 3. Analysis and Results

### 3.1. Anthropometric and Hematological Data

#### 3.1.1. Anthropometric Measurements and Hemoglobin Levels in Study Children

Every second child was found to be stunted (51.9%) or underweight (49.2%), and every fifth child suffered from moderate or severe wasting (19.0%). According to MUAC z-score measurements 15.3% of children were classified as malnourished. When both criteria WHZ and MUAC are compulsorily used (*n* = 306), then merely 9.5% would be classified as malnourished, whereas a complementary use of both cutoffs results in an overall prevalence of 24.9%. The Spearman correlation detected a moderate positive association between MUAC z-scores and wasting in children with a correlation coefficient of 0.693 ** (Table 1).

Between boys and girls no significant differences concerning anthropometric measurements were detected (except for physiologically indicated differences in weight and MUAC (cm)). Children (*n* = 307) were aged on average 22.5 ± 9.5 months at baseline with no significant difference in the age distribution between both sexes. There was a tendency that girls suffered slightly more from any form of malnutrition according to levels of HAZ, WAZ and WHZ; whereas boys were slightly higher represented in the adequate z-score levels (Table 1).

The proportion of children affected by moderate or severe stunting showed an increase with age, this increase was shown to be highly significant (*p* = 0.000), between the youngest age group versus the older age groups, respectively. Especially the step from the first to the second age group bears a massive increase in the stunting prevalence from 27.8% to 48.6%. Similarly, the proportion of children suffering from moderate or severe underweight increased by age, however the decrease in mean WAZ scores remains non-significant (*p* = 0.055). For wasting the youngest age group of children aged 6–11 months holds the highest proportion of wasted children, then during the second year of life the lowest wasting prevalence was found with subsequent increase in the upper age categories. Concurrent wasting and stunting affected 12.4% (*n* = 38; Table 2).

According to MUAC z-score measurements there was a steady increase in the prevalence of malnutrition by age (Table 2).

Concerning hemoglobin levels 93.8% of children were affected by anemia, hereof the majority (68.7%) suffered from moderate anemia and 4.6% were severely anemic. Gender of the child did not affect anemia prevalence (Table 3).

The Spearman correlation detected a weak positive association between the hemoglobin of children with increasing age with a correlation coefficient of 0.312**. Substantiating this the mean Hb levels was found to increase per age group significantly (*p* = 0.000). Similarly children aged below two years predominantly suffer from moderate anemia, this proportion shifts to an increase of mild or non-anemia in the older age groups. The proportion of severe anemia remains quite stable across the different age groups (Table 3).

Scores for HAZ, BAZ and MUAC showed no correlation with Hb levels. Scores for WAZ, WHZ and Hb showed a very weak association with a correlation coefficient of 0.154** and 0.152**, respectively. The overall comparison of mean Hb levels between children suffering from any anthropometric failure (HAZ, WAZ and WHZ) and other children revealed no significant difference.

#### 3.1.2. The Composite Index of Anthropometric Failure (CIAF) Versus Conventional Indices for the Assessment of Undernutrition

Stunting is an indicator for chronic undernutrition, which manifests in linear growth failure; wasting is an indicator for acute undernutrition with loss of lean and fat tissue; underweight is an indicator reflecting both acute and chronic undernutrition [32]. These indices characterize different facets of undernutrition, however are not mutually exclusive categories, rather a child can be affected by several of these indices at the same time. Thus, the conventional indices fail to assess the overall prevalence of undernutrition in a group. The composite index of anthropometric failure (CIAF) better reflects the overall magnitude of undernutrition and also identifies children without any form of anthropometric failure [33,34]. Moreover CIAF detects multiple anthropometric failures, which is of relevance as the concurrence of wasting with stunting, predicts a high risk for child mortality [35,36].

By applying CIAF criteria versus the conventional indices a higher proportion of children was identified as undernourished (61.6%), as opposed to merely 51.9%, 49.2% and 19.0% of children being classified as stunted, underweight and wasted by conventional indices, respectively. Thus CIAF could recognize 9.7%, 12.4% and 42.6% more children than HAZ, WAZ and WHZ respectively. Age specific differences (6–23 months versus ≥24 months) in regard to single, dual, multiple or total anthropometric failure could be detected in the category of multiple (*p* = 0.030) and total anthropometric failure (*p* = 0.008) by chi-square test. No significant gender specific differences were found (Table 4).

A MUAC (<11.5 cm) is not only identifying children suffering from severe acute malnutrition, but also is indicative for an increased risk of mortality [37]. CIAF could identify 14.2% more children being affected by single or combined forms of wasting than MUAC (Table 5). Hence, application of combined undernutrition indicators appeared clearly superior to searching for wasted children by merely measuring MUAC.

#### 3.1.3. Nutrition-Specific and Sensitive Drivers of Hemoglobin Levels ≥10 g/dL

The determinants age, underweight, child’s morbidity history, maternal hemoglobin level, as well as the count of food groups were assessed to be significant independent key drivers determining Hb ≥ 10 g/dL (Table 6). Linear regression showed an increase of 0.038 g/dL in hemoglobin level per month of age, thus an age-dependent increase of 0.46 g/dL was expected per year of life (*p* = 0.000).

Details of selected anthropometric, socio-economic, morbidity and agricultural background characteristics by hemoglobin levels are depicted in Table 7.

The mean count of food groups consumed during the previous 24-h before the interview (*n* = 264), by children having an Hb level ≥ 10.0 g/dL was significantly higher (*p* = 0.007) as opposed to children with lower baseline Hb levels, which is also reflected in the overall range of food group consumption (2–6 FG versus 0–6 FG), respectively. Similarly, a higher percentage of children suffering from moderate and severe forms of anemia (*n* = 198), failed the minimum dietary diversity indicator [38] (≥4 FG, excluding breast milk substitutes in case of non-breastfed children *n* = 1) as compared to children with mild or no anemia (*n* = 66; 37.4% versus 24.2%). In regard to different types of food groups consumed: the consumption of “starchy foods” (*p* = 0.008), “legumes” (*p* = 0.003) and “chicken egg” (*p* = 0.016) was significantly increased amongst children in the upper Hb category. Equally the consumption of “cow milk” (34.3% vs. 23.1%) or “other fruits and vegetables” (68.2% vs. 54.5%) was more prevalent for children with Hb levels ≥ 10.0 g/dL. The consumption rates of “vitamin A rich fruits and vegetables” (indicated by 26.1% of *n* = 264 children), and “flesh foods” (indicated by 9.7% of *n* = 269 children) were comparable (less than 5% difference) between both groups. When queried for the usual daily feeding routine (*n* = 273), then 5.8% of mothers stated to feed nothing than breast milk/formula to their child even though aged above 6 months, altogether these children were presented in the lower Hb category. This is equal to 6.1% of mothers (*n* = 264) indicating zero food groups during the previous 24-h. The presence of non-eaters in the lower Hb-group explains the significant higher consumption rates of common foods like “starchy foods” or “legumes” amongst children with more satisfying Hb levels.

When seeking to identify predictors for a dietary diversity consisting of four or more food groups (out of possible seven food groups) by log binary regression (*n* = 83/241), then “child’s age at baseline” (*p* = 0.002) and “the availability of any land for cultivation” (*p* = 0.014) were independent determinants of the count of food groups consumed; whereas maternal educational level, HH available cash income or parity were not related to the count of food groups consumed. In turn “any land cultivation” was significantly correlated with cash income groups (r = 0.158**, *p* = 0.008).

All in all, these findings suggest a diversified food intake to promote higher Hb levels, and that a timely introduction of adequate complementary feeding is decisive, moreover agricultural incentives seem to promote a more diversified food intake, possibly through increased cash income. The importance of a diversified diet is also reflected in the regression model (Table 6), which predicts Hb levels ≥ 10 g/dL in children when: aged above two years, falling less often sick, having a lesser tendency of suffering from underweight, living in households where mothers offer diversified meals or having mothers being less likely to suffer from anemia. Hence, a useful leverage to overcome anemia amongst Santal Adivasi children in Birbhum appears to diversify meals (which commonly consist of rice, potato curries and little dhal) and—very importantly—to guide mothers for early and adequate start of complementary feedings (commonly biscuits, rice and potato are offered) to prevent an increasing anemia prevalence in the first two years of life where average Hb drops from 8.8 (*n* = 54) to 8.7 g/dL (*n* = 109)—compared to 9.4 g/dL in the third year of life (*n* = 124) and 10.0 g/dL in the fourth year of life (*n* = 20; Table 3).

#### 3.1.4. Hemoglobin Levels by Maternal Educational Level or Household Wealth

When investigating anemia prevalence by the two main groups of available cash income per month, then it is becoming apparent that the incidence of moderate or severe anemia is declining, and the proportion of mild or non-anemia rising by increasing income. The chi-square test confirmed this observation (*p* = 0.023; Table 7). However no correlation could be detected between the HH available cash income groups and the Hb levels of children; similarly the comparison of mean Hb levels amongst the two main groups of available cash income showed no significant difference. Children suffering from any anthropometric failure distributed equally on the two cash income groups, thus differences in available cash income was not related to the incidence of undernutrition in this study. Moreover, anemia in the child and educational level of the mother showed no interrelation.

These findings are substantiated by the forward regression model (Table 6), as well as the enter regression model with adjusted odds ratios (data not shown), both testing for positive drivers of child hemoglobin levels Hb ≥ 10 g/dL. Altogether in both models the same five predictors were found to be significant out of initially selected 11 predictors (see methods section), with cash income (*p* = 0.482) or educational level (*p* = 0.879) not contributing significantly.

### 3.2. Socio-Demographic Information of Caretakers of Study Children

#### 3.2.1. Background Information

94.2% (*n* = 274) of all caretakers participating in the baseline survey (*n* = 291) belonged to the Santal community (others to Konra tribe).

#### 3.2.2. Household Characteristics

The type of household (HH) structure (*n* = 291) was balanced between nuclear family (49.1%) and an extended form of HH structure, where father, mother and children live together with other relatives under one roof (50.9%). Data suggests that extended families are less vulnerable with commonly two persons earning an income and a lesser (but still very high) share of available income being spent on food: 67.6 percent (*n* = 148) rather than 73.2 percent (*n* = 143). Accordingly, extended families show bigger likelihood of being able to afford seeing a doctor in case of need and having two cooked meals a day throughout the year. The range of HH members was between three and 16 persons (on average five persons). The number of own children in one HH ranged from one to five (on average 1.7 ± 0.8 children, whereby Santal women aged 30 years and older (*n* = 32) commonly have three children). The type of dwelling (*n* = 291) was in the majority of cases (81.4%) a mud house with straw or tin roof, 17.2% afforded a semi-pukka house, here either the roof or the walls are made from pukka materials (bricks, cement, timber, etc.), but merely 1.4% lived in a brick house.

The head of the HH (*n* = 289) was in 91.3% the husband, parents in law or elders, and in merely 8.7% the wife alone or together with husband. Household assets (*n* = 265) available were in the majority of cases electricity (86.4%), light bulb (84.5%), a bicycle (87.9%) or a mobile phone (64.9%). Less than half of all HH (43.0%) had a fan, one quarter (25.7%) a TV, less than one fifth (17.4%) had a toilet and a minority of HH had a scooter (5.7%), a private tubewell (4.9%), a radio (1.9%), a gas cooker (1.9%) or a fridge (0.8%). In 76.6% of HH (*n* = 290) at least one person can read a simple message.

#### 3.2.3. Aspects of Food Security

The majority (95.9%) of all families (*n* = 291) pursued casual labor during the last 12 months. Working on big landowners land to cultivate rice is the most commonly done casual work amongst Santal families surrounding Bolpur (amounting to 81.7% of all casual works, *n* = 263). Moreover, during the last 12 months, half (47.8%) of all families (*n* = 291) had any family member being engaged in self-employment agriculture, whereby 38.0% of families (*n* = 137) fully depend on leased land and another 21.2% cultivate leased land additionally to own plots, which are insufficient to feed the family. Self-employment in the non-agricultural sector was of minor importance (3.4%): besides running a grocery store (45.5%, *n* = 11), driving a tractor plays a role with 27.3%. A small minority of 1.7% of 290 families has been regularly employed. Further, more than half (66.7%) of all families (*n* = 287) were beneficiaries of the “Mahatma Gandhi National Rural Employment Guarantee Act” (MGNREGA), guaranteeing 100 days of employment to every family in possession of a job card and nearly half of surveyed Santal families (47%, *n* = 285) state that the MGNREGA significantly increases available cash income.

Therewith, MGNREGA replenishes rather meager incomes, which amounted to 24.89 rupees per person and day, falling below the poverty line of 1.25 USD (which was still valid in February 2015 when baseline survey was conducted—from October 2015 1.90 USD was set as poverty line [39]) by a tremendous 68%.

Confirming agricultural dependence, 80.6% of families (*n* = 288) rated casual labor as main source of income, followed by self-employment agriculture (21.5%). Minor roles play self-employment activities in the non-agricultural sector (1.4%) and regular employments (0.7%). 2.1% classified the MGNREGA as the main source of income. Hereby of 240 families 20.8% described the MGNREGA to increase the family income, and 21.7% stated that the MGNREGA increased bank savings. One third (30.8%) complained not to be in the possession of a job card (be it due to limited knowledge on the Act or political slants, for instance), and that only one family member is in charge of it (2.1%). Other statements by less than 10% were “wages were not provided in time/payment is late”, “more jobs are needed in dry season when no money and no other work is available” and by less than 5%: “improves income in off-season, when less work is available”, “the number of working days provided were insufficient”/“even worked less days under MGNREGA than without” (individuals may abandon other job options in the time of MGNREGA, which in the retrospect would have been more profitable), “improves financial situation of family”, “irregular work offer, thus not so helpful” and “all MGNREGA is spend on food (no increase of savings)”. Overall, the MGNREGA seems to support Santal families well and gives some financial security to them. Knowledge and legal rights related with the MGNREGA may well be further extended to benefit even more from this valuable government scheme, where work should be provided on demand in times of need within 15 days of announcement and each rural household would be eligible to hold a job card representing all adult household members [40].

The majority (82.3%) of all HH (*n* = 288) stated the income to be sufficient to cook two warm meals a day throughout the year. Still almost every fifth family (17.7%) faces food scarcity in particular in autumn (October and November) and some earlier in the year (starting from July in the rainy season).

How do Santal Adivasi families cope when they fall short of food? Half of households (50.2%, *n* = 283) dispose over some savings, which are usually kept untouched (39.3%, *n* = 135) and finally used to buy food in times of need (46.7%)—other purposes to expense savings are medical treatments (34.8%), investments (14.8%) and buying hygiene articles or stationeries (9.6%), as well as paying the child’s education (5.9%) and buying HH assets (5.9%). Further coping strategies comprise searching for casual work (64.1%, *n* = 142), taking credit to buy food (50.0%) and asking neighbors and relatives for help (35.9%). Moreover, reducing the number of meals (18.3%), reducing the size of meals (7.7%), mother taking less food (5.6%) and selling of belongings (5.6%) play a role. One respondent told to cut grass to sell it and another used income generated from driving a tractor owned by the family—however, these are exceptions and getting into debt is a common consequence of food shortage.

This high vulnerability is underlined by the fact that Santal families use to spend a large proportion of their available income on food. 98.3 percent (*n* = 291) of HH state that most part of monthly available cash income is spent on food—besides, available income is commonly spent on medical purposes (92.1%) and hygiene articles/stationeries (50.5%), as well as HH assets like TV/bicycle (21.3%)%), investments to buy agricultural assets or to run a shop (21.0%), child’s education (10.3%), liquor (5.2%) or cloths (1.4%). On an average, extended Santal families (*n* = 148) spend 67.6% of the available monthly cash income on food, while nuclear families (*n* = 143) need to expense 73.2% of available income to cover food expenses.

The person deciding how to spend HH income or savings (*n* = 280) is mostly the husband alone, parents or parents/brother-in-law or elder (73.7%), while wife alone or wife together with husband decides in 26.4% of HHs.

#### 3.2.4. Hygiene Habits

Of 291 mothers the majority (84.9%) usually are washing their hands with water only, merely every tenth woman (10.7%) is using soap and 4.5% reported to sometimes wash their hands with water only and sometimes used soap in addition. Before preparing food (*n* = 291) 99.3% normally are washing their hands, before eating (*n* = 291) 99.7% usually are washing their hands and all women (100.0%) indicated to wash their hands after using the toilet. Hence, while hand washing is a common habit amongst Santal Adivsi communities surrounding Bolpur, usage of soap is suboptimal, aggravating the spread of diarrheal disease—the second most important cause of under-five-deaths in India with 22.2% [41].

Of 287 women 40.4% usually offer boiled water to their child for drinking, with the majority of women (59.6%) reporting not to offer boiled water to their child. Food in the HH (*n* = 289) was stored covered in 96.5% of cases. Animals of the HH (*n* = 291) can reach the kitchen or dining place in every second HH (47.4%).

Of 288 women the majority had to go for toilet outside the house in nature (88.2%), every tenth woman reported to have a family toilet (9.4%), 3.1% could use a community toilet and in 1.4% the HH toilet was under construction. The fact that open defecation is widely practiced in surveyed Santal communities favors the spread of anemia, which is (besides insufficient nutrient intakes) known to be caused by hookworms, which are transmitted through stools [42], while potential hosts walk barefoot (a common habit in Santal villages).

For bathing the youngest child (*n* = 283), the majority of women (82.3%) used tube well water, 12.0% used pond water, 3.9% reported to use sometimes water from the tube and sometimes from the pond, one woman (0.4%) reported to use boiled water, and one woman (0.4%) to use water from the well. Throughout the year (*n* = 289), 70.2% of women had access to clean water from a tube well or well, almost every fifth woman (18.7%) reported only to have sometimes access, in 8.3% most time of the year there is not enough water coming from the well and in 2.8% of cases no clean water is available at any time.

General access to water (*n* = 282) is getting scarce during summer time (culminating in May and June) in one third of cases (32.6%), whereas the majority (67.4%) reported no water scarcity throughout the year. Consequences (*n* = 16) of the water scarcity include less tube well water for drinking (12.5%), less pond water for washing (68.8%) and a longer walk to water sources (18.8%). Above all, inadequate water access is known to fuel health risks [43] and finally child morbidity—as well as mortality, in a setting of widespread undernutrition.

#### 3.2.5. Morbidity Pattern and Visible Signs of Malnutrition of Children and Their Mothers

Of 302 children, every fourth (23.8%) was diagnosed with paleness of nails or the skin (Table 8). When cross-tabulated with the severity of anemia the paleness of the skin detected 24.4% of all children (*n* = 275) being anemic according to their hemoglobin level. Non-anemic children (*n* = 17) were adequately not diagnosed with signs of paleness, except *n* = 1. This data suggests that checking for paleness of nails is an adequate, yet insufficient screening approach to recognize anemic children (where 70% of anemic children in the sample would have remained unrecognized if blood testing had not been done).

Other visible signs of malnutrition in children diagnosed by doctors were dry eyes (40.0%), dental disorders (9.6%), Bitot’s spot (1.2%), perlèche (3.3%), skin lesion or rough skin (1.3%).

Heart defects were detected in 11 (3.6%) of all children (*n* = 307), hereof seven were guided for surgery and the remaining four were monitored. One child was suffering from cerebral palsy (*n* = 1).

At the baseline medical checkup more than half (63.0%) of all children (*n* = 303) were found being affected by cold or cough, nearly one fourth (24.1%) suffered from fever as indicated by the caretaker during the last week or diagnosed by doctor and almost every fifth child (17.2%) was reported by the caretaker to have suffered from diarrhea (defined as watery stool at least once a day, or soft stool three times a day). 12.8% of children were referred to hospital to receive further diagnostic and treatment. Similarly according to reports from the baseline survey about two thirds (64.5%) of all children (*n* = 282) have suffered from cold or cough, one third (33.9%) of all children (*n* = 280) from fever and every tenth (11.7%) child of all children (*n* = 281) from diarrhea, during the last two weeks prior to the interview.

Morbidity was highest in children aged 12–23 months and declined thereafter (Figure 1). When comparing any morbidity prevalence by age (6–23 months versus ≥24 months), the difference was significant (*p* = 0.036), hereby the discrepancy is rooted in the prevalence of respiratory diseases (*p* = 0.042), whereas fever or diarrhea prevalence was similar between both groups. Similarly anemia prevalence declined by age after reaching the second year of life (Table 3), and was related to morbidity as well (Figure 2, Table 6 and Table 7).

Still, when considering age-related trends of morbidity or morbidity prevalence by anemia status the small sample size for the oldest age group (36–39 months) and the category “severe anemia” has to be taken into account. In particular for Figure 2 a steady rise of morbidity rates by increasing severity of anemia was presumed. No significant differences in morbidity by sex were found.

#### 3.2.6. Health Access and Health Seeking Behavior

Despite the depicted tense undernutrition situation, the vast majority of mothers (87.2%, *n* = 282) perceived her child as “active” (playing, laughing and speaking). Merely every tenth mother said her child was “not much active” (does not like to play much and seems sleepy; 10.3%) and 2.5% described the condition of her child as “sick again and again”.

In case a child falls sick, most mothers do not change the child’s diet in general (72.5%, *n* = 291). However, providing oral rehydration solution (ORS) solution in case of diarrhea is fairly common, with 61.2% of mothers (*n* = 196) having provided this rehydration the last time when the child suffered from diarrhea. Still, more than one third of mothers (37.8%) did nothing special, putting the child at risk to suffer severely from the diarrheal episode. One mother (0.5%) reported to give more fluids, and another mother mentioned that she brought her child to a child specialist (0.5%).

Of 288 HH more than half (67.7%) stated the HH income to be sufficient to see a doctor in case of need—while one third of families remains without capacity to pay a doctor. When mothers perceived sickness in their child (*n* = 286), 99.7% do go to a health facility (government health centers are generally free of costs). Very interestingly, the most frequently consulted person was the village doctor (76.9%, *n* = 285), known as a “quack”—these health providers lack formal health training and treat mainly by experience. Even though villagers perceive them as helpful as they are available nearby the house 24 h a day and prescribe powerful medicines at reasonable costs (quack’s treatment fees were found to be 62% less compared to allopathic doctors [44]), this service needs to be considered with care, as frequent use of antibiotics is known to harm the digestive system, unfavorable to already malnourished children. The second most frequented health institution is government health centers (14.7%, *n* = 285), thirdly government hospitals are consulted (11.2%), further child specialist doctors (8.7%), health centers run privately by an NGO (6.3%) and homeopathic doctors (1.0%).

Government health centers were found to enjoy only partial satisfaction amongst the respondents, even though they are generally in good reach: the distance to the next government health care centre was in the majority of cases “very close” (less than 20 min of walking; 61.9%), or “not much far” (20–40 min by walking; 32.2%), with only 5.5% needing bus or cycle to reach it, and one person (0.3%) not knowing where the next centre was. The services offered by the government health care centre (*n* = 279), were described as “not much helpful” by one third of respondents (27.6%), while half of all women (52.7%) rated the services as “much helpful” and one fifth of women (19.7%) as “more or less alright”. Women that were unsatisfied with the health care (*n* = 71), specified reasons like “doctor does not come/is not in time” (36.6%), “no child specialist doctor always available” (26.8%), “during night closed” (18.3%), “no good quality of provided medicine” (11.3%), “medicine not available but has to be purchased elsewhere “(5.6%) and “no good service” (1.4%).

#### 3.2.7. Access to Food-Home Production and Local Markets

Of *n* = 291 HH, half (52.9%) did not cultivate any land, every fifth family (19.2%) cultivates their own land, 17.9% have only leased land available for cultivation and every tenth (10.0%) HH cultivates their own land and additionally leases land.

Of *n* = 289 families, about half (47.4%) did cultivate rice for their own consumption (while many more depend on agriculture as daily laborers). Hereof, in 55.2% of cases the cultivated rice was sufficient for consumption throughout the year. The time when the stored rice gets scarce (*n* = 57) was mainly reported for the period August to November.

Of 137 surveyed families, 71.5% are solely cultivating rice, 24.1% usually cultivate other crops aside rice and 4.4% sometimes cultivate other crops aside rice. Crops cultivated aside rice (*n* = 36), were mustard (78.4%), potato (32.4%), dal (24.3%), wheat (18.9%), gram (5.4%) and sugar cane (2.7%). Vegetables (*n* = 286) were never cultivated by the majority of cases (89.5%), whereas a minority of 7.0% reported to have the habit of usually cultivating vegetables, 3.1% did so sometimes and one person (0.3%) had grown vegetables only once or twice yet. Of those cultivating vegetables (*n* = 31) at any time, more than half (*n* = 18) had a kitchen garden for that purpose, 12 families cultivated vegetables on the paddy field and one person reported to grow vegetables on another person’s land in terms of shared cropping.

Fruits availability throughout the year (*n* = 285) was limited, hereby 41.8% of all women reported fruits not to be available at all, half (49.1%) at least sometimes had access to fruits and less than every tenth woman (9.1%) reported fruits to be available at any time in the year. Those having access to fruits (*n* = 166), enjoyed having their own fruit tree (63.3%), bought fruits from the market (38.8%) or obtained advantage from village trees that were accessible for everybody to collect ripe fruits (28.7%).

More than half of all families (*n* = 290) owned any livestock (62.1%). Of those having any livestock (*n* = 180), half of HH (55.0%) had a cow, almost every second HH (43.3%) had chicken or goats (39.4%), every fourth HH (25.6%) had ducks, 14.4% had pigs and/or buffalo, respectively, and 1.2% had some sheep. The reason for keeping livestock (*n* = 178), was “mainly for own consumption” (44.9%), “mainly for sale” (22.5%), “both about equal share” (30.9%), “for agricultural works/cultivation” (1.1%) and “for sale and for agriculture” (0.6%).

Of 257 HH, the vast majority (82.5%) felt some price increase of basic food items during the last 12 months, and 4.7% even reported the increase having been too much, whereas a minority of 12.8% had felt no price increase and reported the prices to be more or less stable.

Coping strategies (*n* = 217) were “we bought less quantity” (52.5%), “we borrowed some money to buy food items” (46.1%), “we stopped to buy expensive items” (16.6%) and “we sold some animals to buy food items” (2.3%). These coping strategies finally mean a loss of working capacity (due to worsened food intakes) as well as an intensified dependence on big landowners (who use to lend money), leaving Santal families in no position to claim for higher (agricultural) wages what makes them again more vulnerable to food price increases.

The majority of HH (89.0%) owned a ration card to receive food from the public distribution system (PDS; *n* = 290). The type of ration card (*n* = 251) was by more than every second HH (55.0%) the above poverty line (APL) ration card, and less than half cases (44.6%) the below the poverty line (BPL) ration card, a minority of 0.4% was in charge of both cards.

The items received with this ration cards during the last month were kerosene oil (*n* = 247), rice (*n* = 101), wheat (*n* = 84), sugar (*n* = 69), edible oil (*n* = 5) or soft coke (*n* = 2).

More than every second HH (63.6%) reported to be dissatisfied with the PDS system (*n* = 248). Named reasons (*n* = 156) were “provided quantity always too low” (53.2%), “variety too low” (26.3%), “low quality” (15.4%), “irregular provision” (5.8%), “subsidized prices still too high” (5.1%), “no BPL card (even though needed, therefore APL card not useful)” (5.1%) and by less than 5%: “no ration card at all”, “only husband has ration card” and “HH fears their class change”. Overall, 65.1% of BPL card holders (*n* = 106), but merely 14.4% of APL card holders (*n* = 132), were satisfied with the PDS system, what is understandable seeing that APL HH basically receive kerosene oil only, and are largely excluded from further benefits.

The food provided by the Anganwadi Center (AWC; *n* = 290), dissatisfied 15.5% of mothers, the majority (68.7%) were fully satisfied, 14.8% felt some concerns and one mother (0.3%) was unclear about it as the centre was too far away for her.

Issues criticized (*n* = 85) were “amount insufficient” (43.5%), “poor quality of food” (29.4%), “boring menu” (21.2%), “same food as available at home is provided (mainly rice/potato), no additional value” (15.3%), “varying amount” (7.1%), and by less than 5%: “no fixed time of food distribution”, “inconvenient time of food distribution”, “child is not listed, so does not receive food” or “AWC worker always quarrel, thus they do not cook good food”.

Freshly cooked meals provided by AWCs six days a week have become a reliable source of food amongst Santal families, who are largely happy with this government scheme.

Half (47.7%) of all HH (*n* = 265) indicated mainly to purchase food, and every fifth HH (19.3%) primarily obtained food through their own production (planting, hunting, fishing and collecting) and one third (32.6%) indicated an equal share between purchase and own production. Hence, despite being largely involved in agricultural activities, most Santal families depend on cash income to purchase various food items to feed the family.

The contribution of different measures “to access food” was rated as very important or important in the following descending order: village markets (*n* = 291, 98.2%), the AWC meal provision (*n* = 290, 93.1%), hunting (*n* = 291, 79.3%), income of 100 days work (MGNREGA; *n* = 289, 49.1%), own agricultural production (*n* = 290, 47.9%), food sharing or the exchange of food with neighbors or relatives (*n* = 290, 37.3%), the government support through the PDS system (*n* = 282, 35.1%), own livestock keeping (*n* = 290, 33.8%) and markets at the commercial centers/cities (*n* = 290, 19.3%).

#### 3.2.8. Family Food

Named main characteristics of a healthy family meal (*n* = 291) were rice (98.3%), vegetables (90.4%), dal (64.3%), fish (32.0%), egg (17.2%), potatoes (12.7%), milk (10.0%), leafy vegetables (6.2%) and meat/chicken (2.7%). The quantity of family meals consumed during the last seven days (*n* = 285) was felt adequate in the majority of cases (80.0%). The quality of family food during the last seven days (*n* = 287) was rated as adequate by 70.4% of women. Named factors determining adequacy of a meal (*n* = 204) were an adequate quantity/frequency of meals (64.7%), 6.4% mentioned an adequate quality to be decisive for perceived adequacy of a meal or 2.0% reported that a meal shall be healthy, moreover certain components like vegetables (25.0%), rice (24.0%), dal (21.6%), fish (4.9%), meat (1.0%), potato (1.0%), egg (0.5%) or oil (0.5%) were reported to contribute to adequacy of a meal. Inadequacy of a meal was perceived when the meal size is insufficient (8.2%) or when certain components were missing or in too little quantity like fish (76.5%), dal (41.2%), egg (32.9%), milk (17.6%), vegetables (16.5%), rice (5.9%) or meat (5.9%).

When asked about their understanding of the term food quality (*n* = 124), more than half of women perceived “nutritious food” (54.8%) to be decisive for food quality, less than half (40.3%) named “tasty food” to be decisive for food quality, followed by “fresh foods” (16.9%), “clean hygienic preparation of food, well-boiled food” (10.5%), “less spicy food” (4.0%) and “good food quality/condition of food” (3.2%).

The strongest influence on choice of foods consumed (*n* = 281), was availability (67.3%), price (28.8%), good for health/health aspects (26.7%), preparation time (24.2%), taste (19.2%) and appearance (13.9%). The majority of all women (*n* = 281) thought that it is important to consume vegetables and fruits regularly (92.7%). The reasons for the importance of vegetables or fruit consumption were “good for health/less sick” (71.1%), “contains vitamins” (15.6%), “nutritious/improves nutritional status” (7.1%) and with less than 5% stating: “doctor told”, “contains protein”, “gives strength/energy”, “everybody says”, “side dish”, “parents told”, “increases blood” and “nutritious at lower price/cost”. The majority of all women (*n* = 290) would like to consume fruits or vegetables more often as they presently do (94.1%). The reason for not having yet increased the fruit or vegetable consumption (*n* = 262), were “no money to buy it” (82.1%), “not available nearby the home” (17.2%) and “family members don’t like it much” (5.0%). Altogether these findings suggest the maternal knowledge or awareness on healthy family nutrition to be satisfying and indicate aspects of food security or economic motives as major barrier for absent behavioral change.

Moringa leaves, which grow wildly in Santal villages, (*n* = 291) were consumed in the majority of HH (80.4%). Reasons for not consuming moringa leaves (*n* = 43) were “no tree available” (37.2%), “no good taste” (25.6%), “belief/fear that blood pressure could decrease” (23.3%), “not always available” (14.0%) and “unable to digest” (2.3%). Reasons for consuming moringa leaves (*n* = 234) were as a “side-dish (vegetable)” (79.1%), “only little into sauce” (10.7%), “healthy” (8.5%) and “lowers blood pressure” (6.8%). Of those, who were consuming moringa leaves (*n* = 219), the majority (72.6%) consumed moringa throughout the year, and one third (27.4%) consumed it only seasonally (rainy season). On average the families (*n* = 178) included moringa leaves into their family diet 2.9 ± 3.6 times per month.

## 4. Discussion

### 4.1. Prevalence of Undernutrition

Commonly reported main drivers of a poor child nutrition status are a limited quantity and quality of food due to poverty, a high rate of infectious diseases because of limited access to safe-drinking water, sanitary facilities and health care, and a poor utilization of available services due to low educational level and lack of awareness [45].

WHO 1995 [46] suggested cutoffs to interpret a country’s prevalence of stunting, underweight and wasting. Rates equal to or above 40%, 30% and 15% among children under five years of age are defined as critical, respectively. Data of the State of the World’s Children report by UNICEF in India show the prevalence of these three nutrition indicators to account for 39%, 29% and 15%, in the years 2010–2015 [47].

According to the NFHS-4 2015–16 in India [8] (West Bengal [14]) the prevalence of stunting, underweight, and wasting in children below five years of age were 38.4% (32.5%), 35.7% (31.5%) and 21.0% (20.3%), respectively. For scheduled tribes in West Bengal the prevalence beyond these rates with 37.3% (HAZ), 42.0% (WAZ) and 27.8% (WHZ), respectively [14]. Data of this study on Santal children aged 6–39 months of life at baseline revealed similar or higher rates as compared to the scheduled tribes, thereby exceeding the critical population cutoffs for all three indicators 51.9%, 49.2% and 19.0%.

Compared to findings of the present study, research on Santal children in the Paschim Medinipur District of West Bengal assessed even higher rates for stunting (54.2%), underweight (65.2%) and wasting (20.1%) [48]. Santal children in Birbhum District, West Bengal were found with similar rates of stunting (47.8%), lower rates of underweight (31.1%) and higher rates of wasting (29.6%) [49]. Santal-Munda tribal children in Parganas District, West Bengal were less affected by HAZ and WAZ, in 21.0% and 38.7% of cases, but suffered from higher rates of wasting 32.7% [50]. Santal children in Purulia District, West Bengal showed a much lower prevalence of undernutrition (HAZ (26.3%), WAZ (38.2%) and WHZ (12.7%)) [51]. Among tribal children in Bangladesh the ratios were equally lower with 42%, 28% and 13%, respectively [52].

The prevalence of severe wasting in rural children aged below 5 years in Northern India accounted for 2.2%, as compared to 3.3% in this study [53], or 6.7% for rural West Bengal. The prevalence of severe wasting in this study was low, when compared to higher proportions reported for scheduled tribes in West Bengal (10.1%) [14], or rural children aged below five years of Birbhum District (10.6%) [54].

Altogether this high prevalence of undernutrition substantiates the vulnerability of tribal communities. Children of scheduled tribes are exposed to an increased risk of death under the age of five and suffer from increased rates of anemia, stunting and wasting as compared to non-Adivasis [14].

The age-related trends observed in this study are also partly reflected in the worldwide timing of growth faltering. In this study, stunting prevalence steadily increased by age, with a marked rise in the second year of life. Similarly HAZ patterns show dramatic faltering up to 24 months of age worldwide and increasing stability thereafter. In India the highest decrease in HAZ scores also occurred between the first and second year of life with a continuous but slight decrease thereafter up to the fourth year of a child’s life. In this study underweight showed no significant decrease in scores possibly due to a low sample size, but still increasing prevalence by age. This progressive and slow faltering is substantiated by global data during the first 5 years of age. In this study WHZ faltering had its peak in the youngest age group, similar to global data and data from India suggesting faltering during the first year of child’s life, making these children most susceptible to dying. In this study, the lowest prevalence of WHZ was found in the second year of child’s life, followed by a subsequent increase, whereas global data or data from India suggest a continuous increase in WHZ scores from the first year of child’s life [55].

### 4.2. CIAF Versus Conventional Indices for the Assessment of Undernutrition

In this study single anthropometric failures affected 15.4% of children, whereas dual or multiple failures were identified in 33.7%, or 12.4%, respectively. The overall prevalence of undernutrition according to the CIAF criteria accounted for 61.6% in this study, thereby identified more children as being undernourished than the conventional indices. CIAF prevalence rates of other studies conducted in tribal and rural areas of West Bengal ranged from 43.4% to 73.1% [51,56,57,58,59,60,61,62]. The CIAF classification provides a holistic picture of the overall prevalence of malnutrition in a population and may be more appropriate to be applied in the scope of national programmatic actions than the conventional indices [33].

### 4.3. Use of Weight-for-Height and Mid-Upper Arm Circumference to Diagnose Acute Malnutrition

Grellety and Golden, 2016, examined anthropometric surveys on children aged 6–59 months from 47 countries and found that a minority of children would be diagnosed as malnourished when both criteria WHZ and MUAC are compulsory. The discrepancy in both indices varied by country with the majority of children being classified as malnourished by MUAC and other countries where the majority of malnourished children was diagnosed by WHZ alone. Consequently MUAC and WHZ shall be used both as complementary rather than alternative variables to guide treatment incentives. In the case of India with 1498 subjects having being examined, the majority of children was classified as being moderately or severely malnourished by WHZ < −2 (88.3%), whilst less than half (45.1%) were identified as having a MUAC < 12.5cm. Merely one third (33.4%) of children would have been categorized as being malnourished when both criteria WHZ and MUAC applied compulsory at the same time.

### 4.4. Predictors of Anemia

An anemia prevalence above 40% is classified as a severe public health problem [63]. Still, WHO reference cutoffs for the assessment of anemia [20] have to be considered with care, as possibly overestimating the risk of anemia in children aged below two years [64]. Caste has been identified as independent predictor for childhood anemia even after controlling for adult education and HH wealth, thus comprehensive health actions targeted on scheduled tribes are of need [65].

The anemia prevalence accounted for 58.5% (India) [8], 54.2% (West Bengal) and 68.1% for scheduled tribes in West Bengal [14], as compared to 93.8% in this study. In two rural districts of Karnataka, India, 75.3% of children aged between 12 and 23 months were found to be anemic. Assessed drivers for hemoglobin levels were primarily infant’s iron status, maternal hemoglobin levels, family wealth index, food insecurity score, child’s age and C-reactive protein (CRP) levels or male gender (inversely related) [66]. Equally, low family income was significantly associated with undernutrition in a study on Adivasi in Bangladesh [52], or with anemia in children of North India [67]. In the present study a higher HH available cash income (≥5000 INR/month) was significantly related to lower rates of moderate or severe anemia, but both cash income and gender were not identified as main predictors for Hb levels in the regression analysis and were not significantly related to undernutrition. Still other drivers like child’s age, underweight, the incidence of morbidity (inversely related), maternal Hb levels and count of food groups consumed the day before the interview were significantly related to Hb levels in this study.

Child’s age was detected as one important factor influencing the state of stunting and anemia in the present research. Undernutrition, if it does not lead to death, implies long-lasting numerous physiological changes, contributing to lower linear growth as well as lower lean body mass, thereby enhancing the state of undernutrition itself [68]. These alterations may explain the increased prevalence of undernutrition in the upper age groups as found in this study.

The observed increase in Hb levels after the second year of child’s life can be related to the lower incidence of infections and diseases (Figure 1 and Figure 2), as well as the acquired ability to eat family foods. Thereby a higher variety of foods can be ingested by increasing age, which put them at lower risk of suffering from anemia. This age-related decrease in anemia prevalence is also mirrored by the data of the NFHS-4 for India [8]. An age-dependent increase of Hb has been assessed in the present research (0.038 g/dL per month), which is similar to the increase in rural children aged 12–23 months (0.05 g/dL per month) [66]. Moreover it has to be considered that the iron requirements are greatest in the age range 7–12 months due to rapid growth, which also explains the critical Hb levels at lower age. Given that iron deficiency during the first 1000 days of life—including the time of pregnancy—may imply irreversible cognitive and physical deficits, the first 1000 days framework is suggested for identification, prevention and treatment of iron deficiency [69].

Underweight—an indicator for acute and chronic malnutrition predicted child’s Hb in this study. In a study on Santal children aged 5–12 years, hemoglobin levels were found to be related to chronic undernutrition (stunting z-score values) [70]. Similarly stunting predicted Hb concentration in a study on North Indian children aged 6–30 months [67], or in children younger than 5 years in Nepal and Pakistan [71]. Half of all deaths in children (52.5%) have been identified to be attributable to low weight-for-age, demanding the reduction of malnutrition to become a policy priority [72].

No morbidity on the checkup day or during previous week was associated with 2.5 times greater odds for Hb ≥ 10 g/dL in children, as compared to those having been diagnosed with fever, diarrhea or respiratory infection, respectively.

Undernutrition is related to a depression of the immune system implying frequent infections, in turn infectious diseases aggravate undernutrition through increased metabolism as well as reduced nutrient intake/absorption [45,73]. Likewise anemia is associated with an increased risk of morbidity [74,75] in particular from infectious diseases [63] and mortality [76,77] in children; and morbidity in turn works as a driver for anemia [78].

Maternal hemoglobin was found to be a predictor for children’s Hb profile in this study. The association between maternal and child’s anemia may have multiple causes, influenced by jointly experienced environmental risk factors. Still, while some studies found no association [79,80,81]; others suggest infants born to anemic or iron-deficient mothers being more likely to develop iron-deficiency anemia in the first year of life [82], or hemoglobin levels of exclusively breastfed infants being associated to maternal hemoglobin levels [83]. Equally, children of Nepal and Pakistan having an anemic mother were more prone to anemia [71].

In this study the count of food groups consumed the day before the interview predicted Hb levels. Equally, a study on children aged 6–23 months in South Ethiopia detected HH food insecurity, a poor dietary diversity and inadequate infant and young child feeding (IYCF) practices to be significantly associated with anemia [84]. Even though the etiology of anemia is multifactorial [85], it has been established that iron deficiency is by far the most common cause of anemia [86]. Worldwide more than 40% of anemia cases in children and women are estimated to be a consequence of iron deficiency [87]. The majority (76.4%) of all tribes home to West Bengal suffer from anemia with women and girls being most vulnerable to that condition. When compared to other north-eastern tribes the intake of fruits and vegetables was significantly lower among the Santal community. The prevalence of hemoglobinopathies was very low (1.1%), thus confirming Santals as being the most vulnerable to iron-deficiency anemia, e.g., due to low-bioavailable cereal-based diet and chronic blood loss due to hookworm infestation [88]. Moreover a study on children aged 12–23 months living in New Delhi found the majority to suffer from anemia (79.2%), of which 86%–93% was associated with iron deficiency. Iron status was reported as only nutritional factor being significantly inversely associated with anemia [89]. Equally, Hb levels of North Indian children were related to iron status, as well as folate and vitamin B12 status [67].

There are additional studies on predictors for anemia: an exploration of data of the Reproductive and Child Health-II Survey on children up to six years old, living in the northeastern states of India, revealed the male gender, a low to medium HH living standard, as well as maternal parity increasing the risk of anemia and children of literate mothers had a decreased risk for severe anemia [90]. More than half of a total of 16 surveys including cross-sectional data of different countries (amongst others in Bangladesh) suggest the following predictors for anemia: child’s age, current performance of any breastfeeding (both inversely related), malaria, iron deficiency, stunting, underweight, inflammation, low socioeconomic status and poor sanitation [91]. A study in Tanzania found low birth weight and dietary factors (e.g., low consumption of meat, vegetables and fruits) to predict anemia in children under 5 years [92].

### 4.5. Household (HH) Characteristics and Aspects of Food Security

The HH wealth assessed in this study was restricted to basic needs. The high proportion of families in this study being dependent on casual labor (80.6%) as the main source of income, and half of all families (50.2%) having no savings depict the insecurity of employment faced by most families. The Mahatma Gandhi National Rural Employment Guarantee Act (MGNREGA) is already discussed with its limitations, e.g., that it should provide proportionately more jobs during the agricultural lean season and that wages should be paid in time [93], which have been similarly mentioned by participants of this study. Still, the MGNREGA may serve as a social safety net. The availability of proper food storage possibilities and sanitary facilities remain scarce amongst Adivasi, and the usage of safe drinking water for toddlers has to be reinforced.

### 4.6. Morbidity Rates

According to the NFHS-4 in rural West Bengal [14], the percentage of children under age of five suffering from acute respiratory infections (ARI), fever or diarrhea accounted for 3.7%, 13.3% and 5.9%, respectively. Among children having diarrhea the majority of mothers (62.8%) gave oral rehydration solution (ORS), and 7.4% of mothers reported to give more fluids to their child. The disease prevalence assessed in this study highly exceeds the reported proportions of the NFHS-4. The morbidity exposure of boys and girls was found to be similar. When queried about what they did the last time, when their child suffered from diarrhea, a similar proportion of mothers provided ORS to their children (61.2%), one mother (0.5%) reported to give more fluids.

A survey on tribal children aged below five years in Mysore, India reported dental caries (21.2%), intestinal infections (19.2%) and respiratory infections (21.9%) in every fifth child [94]. In the present study, the prevalence for dental disorders was lower (9.6%), the incidence of intestinal infections similar (17.2%) and the proportion affected by respiratory infections much higher (63.0%). A simple annualization of morbidity rates obtained at one point of time should be avoided, as morbidity rates are highly confounded by age and season [95]. As data of this study were obtained in the end of the winter season, the rate of morbidity is even expected higher for the rainy season, when relating to maternal reports on child health in a study from Bangladesh [96].

No significant differences in morbidity rates could be detected between children suffering from any anthropometric failure versus none. In contrast a study on data of the NFHS-2 showed children suffering from multiple anthropometric failures, being more susceptible to morbidity [34].

According to data of the NHFS-4, West Bengal [14], the fulfillment of the indicator defining the minimum acceptable diet [38] was met by an equally distressing percentage of breastfed Adivasi children (20.2%) as compared to breastfed rural children (19.5%), not explaining the observed discrepancy of higher rates of anemia or lower z-scores amongst Adivasis. As the manifestation of undernutrition is multifaceted, apart from inadequate IYCF practices quality and quantity-wise; a delayed, unqualified (confirmed by high consulting rates of the village doctor “quack” in our sample) or omitted treatment in case of child’s sickness may explain disproportionate higher rates of undernutrition.

### 4.7. Health Access and Health Seeking Behavior

According to literature the tribal status has been identified as a significant risk factor for child mortality even after controlling for wealth. Tribal children are significantly less likely to receive treatment as most tribal mothers face difficulties to access health facilities [97]. Modern medical treatment is frequently accessed not before the disease is considered as serious [98]. In the scope of our long-term study two children died at night due to an infection, and a third child drowned in the village pond when the grandmother was entitled to care for the child in the morning. Altogether the death may have been preventable with increased awareness of severity of sickness and emergency signs in the child by the parents, an improved access to health care, as well as awareness about the risks of passive child care.

### 4.8. Access to Food and Family Food

Santals are reported to be aware about the importance of fruits and vegetables however refrain from affording these luxuries to young children due to high prices, as rice is still considered as the basic need of their children [98]. Similarly in this study the term “nutritious” was put into relation to food quality by more than half of cases (54.8%), and the majority (92.7%) affirmed a regular consumption of vegetables and fruits to be important. Of women 41.8% did not enjoy access to fruits, and of those having access merely 38.8% did purchase them. Aside of the lack of purchasing power, the serving of too small quantities may be the limiting factor to achieve a balanced diet. The WHO/FAO expert group on diet, nutrition and prevention of chronic diseases recommends a minimum of 400 g of fruits and vegetables daily [99,100]. Likewise the Expert Committee of the Indian Council of Medical Research recommends an average consumption of 300 g of vegetables (green leafy vegetables = 50 g, other vegetables = 200 g and roots and tubers = 50 g) and 100 g fruits a day [101]. In India the per capita consumption of vegetables and fruits per annum accounts for 76.1 kg and 11.8 kg for vegetables and fruits, respectively—equal merely to a daily average consumption of 241 g of vegetables and fruits [102]. Mono-crop cultivation (merely rice) was widespread and the cultivation of a kitchen garden uncommon within the target population. Homestead food production programs are discussed as a sustainable solution to enhance food security, as well as to reduce wasting and anemia [103,104]. Upcoming results of the “Food and Agricultural Approaches to Reducing Malnutrition (FAARM) study” will reveal if the creation of homestead productivity gardens including the raising of chickens and adoption of improved dietary and hygiene practices, improves in particular the growth and health of children having benefitted of the program for their full 1000 days [105].

### 4.9. Limitations of the Study

Data presented are cross-sectional data assessed at a single point of time, so changes over time are not reflected and causation limited.

Variant types of anemia with different etiologies could be a major confounding factor for the current study. As anemia may also be a consequence of intestinal parasites, lead poisoning, chronic illness, hereditary diseases or other determinants, which were not assessed in this research.

All 21 villages in the sphere of activity of the cooperating NGO were included, with subsequent cluster-randomized to the respective intervention arms. Consequently no randomization on individual level has been performed, thus the population under study may not reflect the total general population.

Birth certificates may not be in conformity with the real delivery date when retrospectively documented, in case of home childbirths.

Recall bias may still be present even though social workers were trained to minimize this confounding factor commonly applying to studies including self-reporting as primary data source.

Moreover, general limitations in regard to sample size apply to this study, e.g., not being powered for all baseline characteristics presented, as the sample size was powered for the intervention purpose.

## 5. Conclusions

This study affirmed high rates of anemia, undernutrition and morbidity in tribal children aged 6–39 months. Rates of stunting, underweight and wasting highly exceeded the critical population cutoffs for all three indicators. Any anemia affected almost every child. To address identified independent drivers of anemia like child’s age <24 months, low WAZ scores, morbidity, low maternal Hb level and lack of dietary diversification, multi-sectoral programmatic actions comprising the key pillars nutrition, agriculture and health are recommended for timely intervention before the child reaches two years of age.

Strategies to combat childhood anemia and undernutrition at a sustainable level have to assure child’s dietary diversification and an improved access to preventive health care, to tackle both iron deficiency and infection. Disease prevention and the prompt management of infections in toddlers may avert initial weight loss that may preclude adequate linear growth over the long term. Frequent morbidity is a specific driver for undernutrition, and underweight was shown to be associated with anemia in this study.

Further preventive activities have to take place towards improving socio-economic conditions and other broadly discussed circumstances of acute and chronic malnutrition like hygienic and sanitary determinants.

Awareness trainings and kitchen garden programs may increase the availability, access and proper utilization of nutritious foods, thereby addressing poverty-related food insecurity and inadequate iron intakes. Apart from seeking locally available iron-rich resources for improved IYCF practices, the enrichment of home-diets with micronutrient powders may have a significant impact on improving hemoglobin levels and iron status [106,107]. Strategies should equally fight maternal anemia. Access to qualitative education may break the intergenerational cycles of disadvantage, as poverty is about more than income [108]. Maternal education is well established to be related to a reduced risk of undernutrition in poor countries [109]. All preventive and acute-medical actions have to take place in a cultural-sensible way to build trust, only by that a fruitful cooperation and long-term guidance between experts and Adivasis can be established to allow sustainable change.

## Figures and Tables

**Figure 1 ijerph-17-00342-f001:**
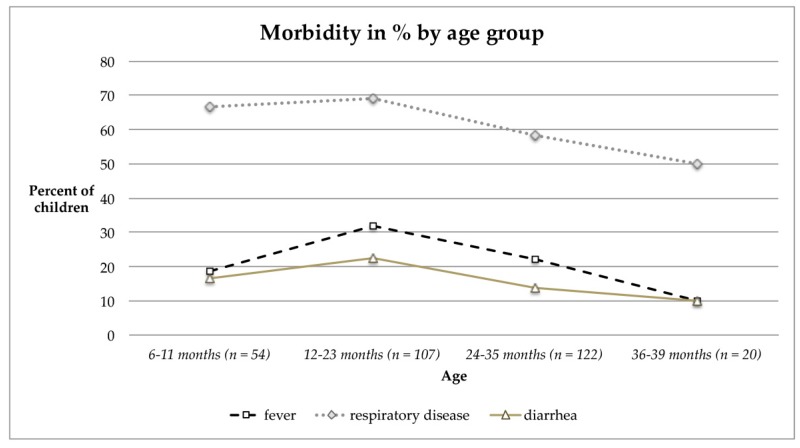
Proportion (%) of children (*n* = 303) by age suffering from fever, respiratory disease or diarrhea during the last week prior/or during medical checkup.

**Figure 2 ijerph-17-00342-f002:**
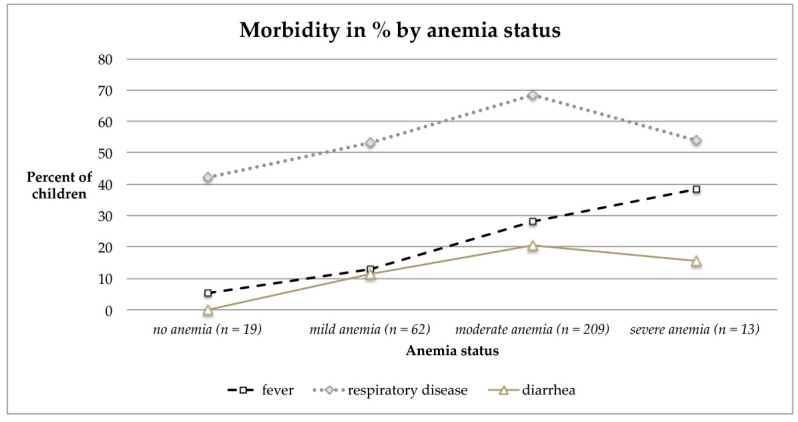
Proportion (%) of children (*n* = 303) by anemia status suffering from fever, respiratory disease or diarrhea during the last week prior/or during medical checkup.

**Table 1 ijerph-17-00342-t001:** Nutritional status of Santal Adivasi children (total) and by gender.

				Mild and Adequate 2 > z-score ≥ −2 MUAC ≥ 12.5 cm	Moderate −2 > z-score ≥ −3 11.5 cm ≤ MUAC < 12.5 cm	Severe −3 > z-score 11.5 cm > MUAC	Total Malnourished
**Total**	***n***		**Mean ± SD/Median, Min/Max**	***n* (%)**	***n* (%)**	***n* (%)**	***n* (%)**
Sex ratio, (*n* = 307)	307		0.872 (872 girls/1000 boys)				
Age (months) ^NN^ ***	307		22.5 ± 9.5/23.0, 6/39				
Weight (kg) ^NN^ *	307		9.1 ± 1.7/9.0, 5.5/14.1				
Height/Length (cm) ^NN^ **	306		78.1 ± 7.5/78.0, 61.5/96.8				
HAZ	306		−2.03 ± 1.11, −5.75/2.00	146 (47.7)	109 (35.6)	50 (16.3)	159 (51.9)
WAZ	307		−1.95 ± 0.98, −4.36/1.44	156 (50.8)	110 (35.8)	41 (13.4)	151 (49.2)
WHZ	306		−1.19 ± 0.93, −3.84/2.51	247 (80.7)	48 (15.7)	10 (3.3)	58 (19.0)
BAZ	306		−0.96 ± 0.96, −3.75/2.87	264 (86.3)	37 (12.1)	4 (1.3)	41 (13.4)
MUAC z-score ^NN^ *	307		−1.09 ± 0.88/−1.02, −3.52/1.69	260 (84.7)	39 (12.7)	8 (2.6)	47 (15.3)
MUAC ^NN^ ** (cm)	307		13.8 ± 1.0/13.8, 11.0/16.8	276 (89.9)	27 (8.8)	4 (1.3)	31 (10.1)
**by Gender**	***n***	***p*-value**	**Mean ± SD, Min/Max**	***n* (%)**	***n* (%)**	***n* (%)**	***n* (%)**
Age (months) Boys	164	0.491	22.1 ± 10.0, 6/39				
Age (months) Girls	143		22.9 ± 8.9, 6/38				
Weight (kg) Boys	164	0.005 **	9.4 ± 1.9, 5.5/14.1				
Weight (kg) Girls	143		8.8 ± 1.5, 5.6/12.9				
Height/Length (cm) Boys	163	0.164	78.6 ± 8.1, 61.5/96.8				
Height/Length (cm) Girls	143		77.4 ± 6.8, 61.5/92.8				
HAZ Boys	163	0.391	−1.98 ± 1.09, −4.50/1.02	85 (52.1)	52 (31.9)	26 (16.0)	78 (47.9)
HAZ Girls	143		−2.09 ± 1.14, −5.75/2.00	61 (42.7)	57 (39.9)	24 (16.8)	81 (56.7)
WAZ Boys	164	0.395	−1.91 ± 0.94, −4.27/1.44	84 (51.2)	61 (37.2)	19 (11.6)	80 (48.8)
WAZ Girls	143		−2.00 ± 1.02, −4.36/0.31	72 (50.3)	49 (34.3)	22 (15.4)	71 (49.7)
WHZ Boys	163	0.866	−1.19 ± 0.96, −3.84/2.51	133 (81.6)	23 (14.1)	6 (3.7)	29 (17.8)
WHZ Girls	143		−1.18 ± 0.89, −3.21/1.18	114 (79.7)	25 (17.5)	4 (2.8)	29 (20.3)
BAZ Boys	163	0.880	−0.96 ± 1.00, −3.75/2.87	138 (84.7)	20 (12.3)	4 (2.5)	24 (14.8)
BAZ Girls	143		−0.95 ± 0.91, −2.97/1.33	126 (88.1)	17 (11.9)	0 (0)	17 (11.9)
MUAC (cm) Boys	164	0.005 **	13.9 ± 1.0, 11.0/16.7	151 (92.1)	10 (6.1)	3 (1.8)	13 (7.9)
MUAC (cm) Girls	143		13.6 ± 1.0, 11.0/16.8	125 (87.4)	17 (11.9)	1 (0.7)	18 (12.6)
MUAC z-score Boys	164	0.784	−1.08 ± 0.90, −3.40/1.69	138 (84.1)	21 (12.8)	5 (3.0)	26 (15.8)
MUAC z-score Girls	143		−1.10 ± 0.86, −3.52/1.20	122 (85.3)	18 (12.6)	3 (2.1)	21 (14.7)

Note: HAZ (1 (0.3%) z-score ≥ 2), WHZ (1 (0.3%) overweight z-score ≥ 2), BAZ (1 (0.3%) z-score ≥ 2). WHZ Boys (1 (0.6%) overweight z-score ≥ 2), BAZ Boys (1 (0.6%) overweight z-score ≥ 2), HAZ Girls (1 (0.7%) z-score ≥ 2). Description of Table 1, Table 2, Table 3, Table 4, Table 5, Table 6, Table 7 and Table 8: Data units presented are age (months), weight (kg), height/length (cm), z-scores (unitless), MUAC (cm) and Hb (g/dL). Values represent mean ± SD/median, min/max or *N* (%) following the indicated appearance (note: median is merely presented for total sample in the case of not normally distributed data according the Kolmogorov–Smirnov (K–S) test). NS non significant. ***** Levels of significance * *p* < 0.05; ** *p* < 0.01; *** *p* < 0.001. For comparison tests the *p*-value is indicated. Levels of significance after adjustment of *p*-values according to Bonferroni *p* ≤ 0.008 *, *p* ≤ 0.0016 **, *p* ≤ 0.00016 *** in the case of four groups to be compared. Significant difference between age groups starting from the youngest: ^a^ I and II; ^b^ II and III; ^c^ III and IV; ^d^ I and IV; ^e^ II and IV; ^f^ I and III. NN not normally distributed tested by K–S Test, the hypothesis regarding the distributional form is rejected with *, **, ***.

**Table 2 ijerph-17-00342-t002:** Nutritional status by age.

				Mild and Adequate 2 > z-score ≥ −2 MUAC ≥ 12.5 cm	Moderate −2 > z-score ≥ −3 11.5 cm ≤ MUAC < 12.5 cm	Severe −3 > z-score 11.5 cm > MUAC	Total Mal-Nourished
Total Age-Related	*n*	*p*-Value	Mean ± SD, Min/Max	*n* (%)	*n* (%)	*n* (%)	*n* (%)
HAZ 6–11 m	54	0.000 *** (a, d, f)	−1.37 ± 1.17, −4.38/2.00	38 (70.4)	12 (22.2)	3 (5.6)	15 (27.8)
HAZ 12–23 m	109		−2.01 ± 1.03, −4.38/1.02	56 (51.4)	35 (32.1)	18 (16.5)	53 (48.6)
HAZ 24–35 m	123		−2.30 ± 1.05, −5.75/0.76	46 (37.4)	50 (40.7)	27 (22.0)	77 (62.7)
HAZ 36–39 m	20		−2.25 ± 1.09, −4.24/−0.10	6 (30.0)	12 (60.0)	2 (10.0)	14 (70.0)
WAZ 6–11 m	54	0.055	−1.71 ± 1.00, −4.09/0.60	29 (53.7)	21 (38.9)	4 (7.4)	25 (46.3)
WAZ 12–23 m	109		−1.87 ± 0.98, −4.09/1.44	61 (56.0)	37 (33.9)	11 (10.1)	48 (44.0)
WAZ 24–35 m	124		−2.10 ± 0.95, −4.36/0.31	58 (46.8)	44 (35.5)	22 (17.7)	66 (53.2)
WAZ 36–39 m	20		−2.11 ± 0.94, −3.69/−0.37	8 (40.0)	8 (40.0)	4 (20.0)	12 (60.0)
WHZ 6–11 m	54	0.964	−1.23 ± 0.99, −3.10/1.18	39 (72.2)	13 (24.1)	2 (3.7)	15 (27.8)
WHZ 12–23 m	109		−1.20 ± 0.96, −3.84/2.51	93 (85.3)	11 (10.1)	4 (3.7)	15 (13.8)
WHZ 24–35 m	123		−1.16 ± 0.89, −3.21/1.01	99 (80.5)	21 (17.1)	3 (2.4)	24 (19.5)
WHZ 36–39 m	20		−1.15 ± 0.87, −3.07/0.46	16 (80.0)	3 (15.0)	1 (5.0)	4 (20.0)
BAZ 6–11 m	54	0.085	−1.26 ± 0.99, −3.2/1.15	40 (74.1)	12 (22.2)	2 (3.7)	14 (25.9)
BAZ 12–23 m	109		−0.89 ± 0.97, −3.75/2.87	100 (91.7)	6 (5.5)	2 (1.8)	8 (7.3)
BAZ 24–35 m	123		−0.89 ± 0.93, −2.89/1.18	107 (87.0)	16 (13.0)	0 (0)	16 (13.0)
BAZ 36–39 m	20		−0.90 ± 0.93, −2.97/0.87	17 (85.0)	3 (15.0)	0 (0)	3 (15.0)
MUAC 6–11 m	54	0.000 *** b: *p* = 0.000352 ** ^Bonfer.^ d: *p* = 0.001409 ** ^Bonfer.^ e: *p* = 0.004542 * ^Bonfer.^ f: *p* = 0.000015 *** ^Bonfer.^	13.3 ± 1.0, 11.0/15.4	45 (83.3)	7 (13.0)	2 (3.7)	9 (16.7)
MUAC 12–23 m	109		13.6 ± 0.9, 11.0/16.7	98 (89.9)	9 (8.3)	2 (1.8)	11 (10.1)
MUAC 24–35 m	124		14.0 ± 1.0, 12.0/16.8	115 (92.7)	9 (7.3)	0 (0.0)	9 (7.3)
MUAC 36–39 m	20		14.3 ± 1.2, 11.7/15.8	18 (90.0)	2 (10.0)	0 (0.0)	2 (10.0)
MUAC_z-score_ 6–11 m	54	0.285	−0.97 ± 0.91, −3.34/0.97	48 (88.9)	4 (7.4)	2 (3.7)	6 (11.1)
MUAC_z-score_ 12–23 m	109		−1.00 ± 0.83, −3.36/1.69	96 (88.1)	11 (10.1)	2 (1.8)	13 (11.9)
MUAC_z-score_ 24–35 m	124		−1.19 ± 0.88, −3.40/1.20	101 (81.5)	20 (16.1)	3 (2.4)	23 (18.5)
MUAC_z-score_ 36–39 m	20		−1.28 ± 1.01, −3.52/−0.01	15 (75.0)	4 (20.0)	1 (5.0)	5 (25.0)

Note: HAZ 6–11 m (1 (1.9%) z-score ≥ 2), WHZ 12–23 m (1 (0.9%) overweight z-score ≥ 2), BAZ 12–23 m (1 (0.9%) overweight z-score ≥ 2).

**Table 3 ijerph-17-00342-t003:** Hemoglobin levels among surveyed Santal children (total), as well as by gender and by age.

				No Anemia Hb ≥ 11.0 g/dL	Mild Hb 10.0–10.9 g/dL	Moderate Hb 7.0–9.9 g/dL	Severe Hb < 7.0 g/dL	Total Anemia
	*n*	*p*-Value	Mean ± SD, Min/Max (g/dL)	*n* (%)	*n* (%)	*n* (%)	*n* (%)	*n* (%)
Hb Total	307	-	9.1 ± 1.3, 5.0/12.7	19 (6.2)	63 (20.5)	211 (68.7)	14 (4.6)	288 (93.8)
Hb Boys	164	0.915	9.1 ± 1.3, 5.6/12.7	10 (6.1)	35 (21.3)	112 (68.3)	7 (4.3)	154 (93.9)
Hb Girls	143		9.1 ± 1.2, 5.0/11.8	9 (6.3)	28 (19.6)	99 (69.2)	7 (4.9)	134 (93.7)
Hb 6–11 m	54	0.000 *** (b, c, d, e, f)	8.8 ± 1.1, 5.8/11.9	2 (3.7)	5 (9.3)	44 (81.5)	3 (5.6)	52 (96.4)
Hb 12–23 m	109		8.7 ± 1.2, 5.0/12.0	2 (1.8)	14 (12.8)	88 (80.7)	5 (4.6)	107 (98.1)
Hb 24–35 m	124		9.4 ± 1.3, 5.0/12.7	11 (8.9)	35 (28.2)	73 (58.9)	5 (4.0)	113 (91.1)
Hb 36–39 m	20		10.0 ± 1.3, 6.6/11.8	4 (20.0)	9 (45.0)	6 (30.0)	1 (5.0)	16 (80.0)

Levels of significance *** *p* < 0.001.

**Table 4 ijerph-17-00342-t004:** Prevalence of undernutrition amongst Santal children according to composite index of anthropometric failure (CIAF).

Group	Description of the Group	Total *n* = 307 *n* (%)	Sex		Age			
			Boys (*n* = 164) *n* (%)	Girls (*n* = 143) *n* (%)	6–11 m (*n* = 54) *n* (%)	12–23 m (*n* = 109) *n* (%)	24–35 m (*n* = 124) *n* (%)	36–47 m (*n* = 20) *n* (%)
A	No anthropometric failure	118 (38.4)	65 (39.6)	53 (37.1)	26 (48.1)	48 (44.0)	39 (31.5)	5 (25.0)
B	Wasting only	2 (0.7)	2 (1.2)	0 (0)	1 (1.9)	1 (0.9)	0 (0)	0 (0)
C	Wasting and underweight	18 (5.9)	12 (7.4)	6 (4.2)	11 (20.4)	3 (2.8)	3 (2.4)	1 (5.0)
D	Wasting, underweight, stunting	38 (12.4)	15 (9.2)	23 (16.1)	3 (5.6)	11 (10.1)	21 (17.1)	3 (15.0)
E	Stunting and underweight	85 (27.8)	46 (28.2)	39 (27.3)	10 (18.5)	30 (27.5)	37 (30.1)	8 (40.0)
F	Stunting only	36 (11.8)	17 (10.4)	19 (13.3)	2 (3.7)	12 (11.0)	19 (15.4)	3 (15.0)
Y	Underweight only	9 (2.9)	6 (3.7)	3 (2.1)	1 (1.9)	4 (3.7)	4 (3.3)	0 (0)
	Total anthropometric failure	189 (61.6)	99 (60.4)	90 (62.9)	28 (51.9)	61 (56.0)	85 (68.5)	15 (75.0)

Note: group A (total *n* = 307), group B, C, D, E, F and Y (total *n* = 306).

**Table 5 ijerph-17-00342-t005:** Distribution of Santal children in mid-upper arm circumference (MUAC) categories across the CIAF.

	CIAF Classification	MUAC Categories *n* = 307		
		Adequate (*n* = 276) MUAC ≥ 12.5 cm *n* (%)	Moderate (*n* = 27) 11.5 cm ≤ MUAC < 12.5 cm *n* (%)	Severe (*n* = 4) 11.5 cm > MUAC *n* (%)
A	No anthropometric failure	116 (42.0)	2 (7.4)	0 (0)
B	Wasting only	2 (0.7)	0 (0)	0 (0)
C	Wasting and underweight	15 (5.5)	2 (7.4)	1 (25.0)
D	Wasting, underweight and stunting	22 (8.0)	14 (51.9)	2 (50.0)
E	Stunting and underweight	77 (28.0)	7 (25.9)	1 (25.0)
F	Stunting only	36 (13.1)	0 (0)	0 (0)
Y	Underweight only	7 (2.5)	2 (7.4)	0 (0)
	Total anthropometric failure	160 (58.0)	25 (92.6)	4 (100.0)

Note: group A (total *n* = 307), group B, C, D, E, F and Y (total *n* = 306).

**Table 6 ijerph-17-00342-t006:** Ratio of odds of having an Hb ≥ 10 g/dL, including 11 predictors (*n* = 46/196, 23.5% prevalence).

Final Model: Forward Wald	Beta	S.E.	Odds Ratio Exp(B)	95% CI	*p*-Value
Age two categories, 1 = ≥24 months	1.613	0.424	5.019	2.185, 11.527	0.000 ***
WAZ	0.517	0.209	1.677	1.113, 2.526	0.013 *
Morbidity history, 1 = no morbidity	0.866	0.398	2.376	1.089, 5.188	0.030 *
Maternal Hb level	0.335	0.163	1.398	1.016, 1.923	0.040 *
Count of food groups consumed during previous 24 h	0.497	0.209	1.644	1.091, 2.476	0.017 *

Levels of significance * *p* < 0.05; *** *p* < 0.001.

**Table 7 ijerph-17-00342-t007:**
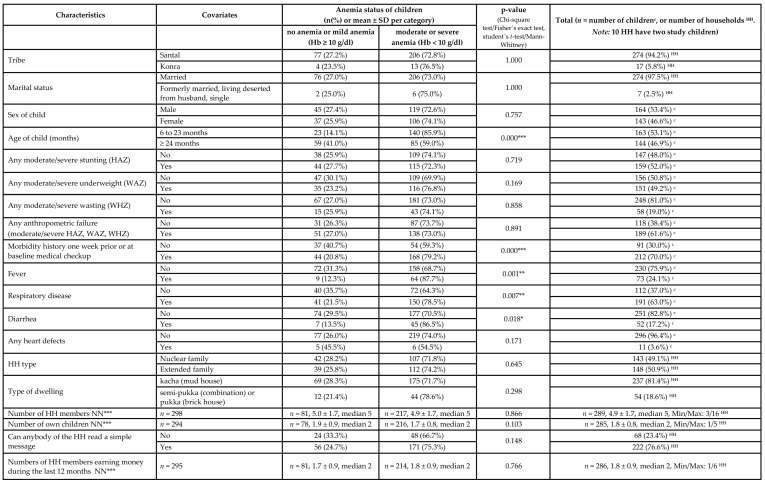
Anemia status of Santal Adivasi children by selected independent socio-economic and child’s/maternal anthropometric characteristics.

Levels of significance * *p* < 0.05; ** *p* < 0.01; *** *p* < 0.001. NN not normally distributed tested by Kolmogorov–Smirnov test, the hypothesis regarding the distributional form is rejected with *, ** and ***. Definition of food groups (FG): starchy foods (grains, roots, tubers), legumes and nuts, dairy products, flesh foods (meat, fish, poultry and organ meats), eggs, vitamin A rich fruits and vegetables and other fruits and vegetables.

**Table 8 ijerph-17-00342-t008:** Morbidity pattern of children and visible signs of malnutrition in mothers and children at baseline medical checkup.

**Visible Signs of Malnutrition at Medical Checkup**	***n* Children**	***n* (%) Children**	***n* Mothers**	***n* (%) Mothers**
Dry eyes/eyeball with wavy structure	275	110 (40.0%)	288	69 (24.0%)
Night blindness	303	0 (0.0%)	288	12 (4.2%)
Bitot’s spot	275	3 (1.2%)	288	22 (7.6%)
Perlèche	303	10 (3.3%)	288	3 (1.0%)
Pale fingernails/paleness of skin	302	72 (23.8%)	287	60 (20.9%)
Skin lesion, rough skin, pigmentation	304	4 (1.3%)	288	74 (25.7%)
Dental disorders/caries	303	29 (9.6%)	288	39 (13.5%)
**Morbidity Pattern at Medical Checkup or during Previous Week**	***n* Children**	***n* (%) Children**	***n* Mothers**	***n* (%) Mothers**
Fever	303	73 (24.1%)	288	23 (8.0%)
Respiratory infections (cold/cough)	303	191 (63.0%)	288	47 (16.3%)
Intestinal infections (diarrhea)	303	52 (17.2%)	288	8 (2.8%)
Referred to hospital	304	39 (12.8%)	288	32 (11.1%)

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
