# Peer review of "Prevalence of Undernutrition and Anemia among Santal Adivasi Children, Birbhum District, West Bengal, India"

_ijerph, 2020, doi:10.3390/ijerph17010342_

Round 1

Reviewer 1 Report

Thank you for the opportunity to review the article “Predictors of moderate and severe anemia (Hb<10g/dl) and prevalence of undernutrition among Santal Adivasi children, Birbhum District, West Bengal, India” for Int. J. Environ. Res. Public Health.

I have some major concerns with the structure and content of the paper and it needs significantly simplified. Below I provide some high-level comments.

The title is rather succinct and self-explanatory; however, I did not feel the content of the paper was in sync with the title of the paper. The paper in its current form is rather lengthy with too many tables and too much information for one paper. It looks more like an extensive report on the study survey. It needs a significant revision and simplification.

Survey Methods and Measures

Have a title maybe Household Survey or just Survey and briefly describe the study survey: who did it, when, why and what kind of information was collected. You might want to merge Village selection, Medical Checkup, and Socio-economic and demographic information briefly and succinctly. Importantly, please provide information on whether random sampling was used, if yes then how. Did you have the denominator of all households, if yes, were then households randomly selected? If not please mention this and then also in the limitation section that the findings are not generalizable for this population. My concern is about the high ratio of response rate >98%. This is unusual and begs more scrutiny as to how it was achieved. Brief info on this would be appreciated.

I would recommend having the following heading and subheading:

Measures

Outcome variable

Predictors or Covariates

Analysis

Under outcome variables, list all your outcomes for anemia and undernutrition and report how each was measured and your rationale and decision what cut point to use to dichotomize them.

Under Predictors (as the title suggest) or covariates subheading, report that the list of predictors is informed by the UNICEF conceptual framework. Provide brief info on the framework and list of predictors. Then reports each predictor that was available from your survey and how it was measured. For example, mother’s education (no formal education as referent vs. any formal education), etc. Note, for each categorical variable inform readers, which category you used as a reference/referent.

For the analysis section given that you do. Not have a comparison group per se, first report that you will conduct descriptive analysis and will report the descriptive in terms of ratios and means and their standard errors for categorical and continuous variables respectively.

Since all your outcomes are binary, just report that you will be using logistic regression and will report unadjusted and adjusted ORs and their 95% CI. No need to explain the difference between RR and OR for rare and common outcomes and their interpretation. Your audience will know this. Using an unadjusted odds ratios will also substitute your bivariate stats. In terms of reporting the results in the table, I would recommend reporting the ratios for each outcome (in the columns) by each predictor (in rows) as the first column, unadjusted OR and 95%CI in the second column and adjusted OR and 95%CI in the third column.

For all your models, naturally, I would also recommend keeping all the predictors in the regression models as long as there is no multicollinearity between the selected variables. You have a compelling reason informed by the UNICEF conceptual model to have them in your models. Hence, I believe there is no need to leave them out from the models even if they are not significantly associated with your study outcomes. Based on your findings, you can then compare these nonsignificant associations to findings from previous research that showed that they were significant. For example, a mother’s education might be powerful predictor in other studies maybe but not in your study sample. You can then speculate why it is so-this is just an example. For the structure of the table, you can put all predictors in the rows and have three columns for outcomes, first for ratio, second of adjusted OR in the columns the ratios of each outcome by each predictor.

Having said this, your results should reflect this analytical approach. First, describing your study sample by all the list of predictors and then reporting the results from your unadjusted and adjusted logistics regressions by comparing the association at the baseline (with no adjustment) and independent association after adjusting for all covariates (from the UNICEF framework). You could even say for example, that out of 15 or 20 potential predictors as per UNCIEF concept in your study, you found only n predictors to be significantly associated with outcome 1 and z predictors with outcome 2 etc.

Again, think about the title of your paper, readers are expecting to learn what predicts undernutrition in your study sample based on multivariable regression analysis.

Finally, limitation section needs to be beefed up. There are many limitations into this study including, small sample, multiple comparison, non-generalizability (if non-randomization was used), recall bias, validity of measures (I would not buy the cash income would be accurately reported by households, or not sure if every single child had a birth certificate, etc.), etc. These limitations are all normal to have, especially in studies involving primary data collection.

Author Response

Dear Reviewer 1,

thank you for your constructive Feedback.

Please see attachment for my comments.

Warm Regards

Reviewer 2 Report

This study was designed to assess the living conditions of Santal Adivasis, the extent of child undernutrition (conventional and CIAF classification) as well as the burden of anemia in children and its nutrition specific and sensitive drivers. The research survey was conducted in 21 Santal villages, Birbhum District, West Bengal, India in 2015. An overall 307 children (aged 6-39 months) and their mothers (n=288) were assessed for their hemoglobin (Hb) levels and anthropometric indices such as height/length, weight and MUAC. Moreover, socio-demographic household characteristics were surveyed. The study confirmed Adivasi children lagging behind national average with a high prevalence of undernutrition (HAZ 51.9%, WAZ 49.2%, WHZ 19.0%, CIAF 61.6%) and of moderate and severe anemia (73.3% altogether). In order to address identified drivers of anemia, namely child´s age<24 months, low WAZ scores, high morbidity tendency, low maternal Hb level, and lack of dietary diversification were identified as factors, thereby warrant intervention to break the intergenerational circle of undernutrition .

In general, the manuscript is well written. This study is creative and has substantial contribution to the current knowledge. However some minor revision should be made on this manuscript before publication.  

My comments  are as follows:   1) There are many types of anemia with different etiologies, and these variant types of anemia can be further classified into megaloblastic, normocytic and microcytic. For microcytic anemia, the main etiologies may be iron–deficiency (IDA) (major), thalassemia (major) or lead intoxication (minor). Variant types of anemia with different etiologies could be a major confounding factor for the current study. However, the types of anemia in children are not mentioned in the current study. The condition need to be clarified, otherwise this point should be listed in the limitation of the study. 2)  There are too many tables, which can be integrated together to provide a focus for the readers. Also, different categories (gender, age, …) can be put together for a comparison of each other (list P value for significance). For example, Table 4 and 5 can be integrated to one table (title: Total; subtitle: Boys and Girls; another title: Age; subtitle: 6-11m, 12-23m, 24-35m,36-39m ). Table 1 and 2 can be integrated to one table (title: Age, Weight, Height, HAZ, ….; subtitle: Age Boys, Age Girls, Weight Boys, Weight Girls, Height Boys, Height Girls, HAZ Boys, HAZ Girls…)

Author Response

Dear Reviewer 2,

Thank you for your constructive feedback.

Please see the attachment for my comments.

Warm regards.

Reviewer 3 Report

Thank you very much for allowing me to review the original article "Predictors of moderate and severe anemia (Hb <10g / dl) and prevalence of undernutrition among Santal Adivasi children, Birbhum District, West Bengal, India" (ijerph-646338).
This is a study on children belonging to India's indigenous population, these children reside in rural areas and with very low income. Being under-five-mortality rates among 55 Adivasis are by 25% increased as compared to non-Adivasi children aged younger than five years. This is a comprehensive study of the status of children based on anemia, with special attention to their nutritional status.

Summary:
Line 27, clarify the acronym MUAC.
Line 28, clarifies the acronym HAZ, WAZ, WHZ and CIAF.
Line 30, concept of anemia, indicate which criterion has been used to identify 73.3% of children with anemia.
Introduction:
It is well posed and explains the situation necessary to understand the work. But the goal "One major objective of this research work was to define nutrition-specific and sensitive 69 drivers of anemia and to elucidate the burden of undernutrition in Santal children." it is not clear, it should be rewritten indicating the main objective and the secondary objectives clearly.
Materials and Methods:
Line 73, clarify if it is a prospective or retrospective longitudinal design. it is not clear "in the scope of a longitudinal feeding trial". Subsequently (line 79) says that it starts in 2015, does it actively start in 2015? Or have they collected the data in 2015?
It is not clear if it is a traversal, since in this article they only use the baseline data. Please clarify.
Line 91 What questionnaires have been used, are they validated?
Line 102, 307 children are studied, on which resident population, what is the participation rate?
Results:
Table 1, 2, 3.- Use one or two decimal places but in all the same, be careful there are some data with three. The tables must be self-explanatory, at the bottom of the table you must indicate the meaning of the acronyms and also the units used. Comparison tests should indicate that they compare. It should also indicate which are mean and SD and which are Medium.
Why are the 6-11m, 12-23m, 24-35m and 35-39m groups used?
Table 4-10 is very interesting.

Discussion:
It is very well structured.

Author Response

Dear Reviewer 3,

Thank you for your constructive feedback.

Please see attachment for my comments.

Warm regards.
